# The Glasgow Prognostic Score Predicts Survival Outcomes in Neuroendocrine Neoplasms of the Gastro–Entero–Pancreatic (GEP-NEN) System

**DOI:** 10.3390/cancers14215465

**Published:** 2022-11-07

**Authors:** Niklas Gebauer, Maria Ziehm, Judith Gebauer, Armin Riecke, Sebastian Meyhöfer, Birte Kulemann, Nikolas von Bubnoff, Konrad Steinestel, Arthur Bauer, Hanno M. Witte

**Affiliations:** 1University Cancer Center Schleswig-Holstein (UCCSH), University Hospital of Schleswig-Holstein, Campus Lübeck, Ratzeburger Allee 160, 23538 Lübeck, Germany; 2Department of Hematology and Oncology, University Hospital of Schleswig-Holstein, Campus Lübeck, Ratzeburger Allee 160, 23538 Lübeck, Germany; 3Department of Internal Medicine, University Hospital of Schleswig-Holstein, Campus Lübeck, Ratzeburger Allee 160, 23538 Lübeck, Germany; 4Department of Hematology and Oncology, German Armed Forces Hospital Ulm, Oberer Eselsberg 40, 89081 Ulm, Germany; 5Department of Surgery, University Hospital of Schleswig-Holstein, Campus Lübeck, Ratzeburger Allee 160, 23538 Lübeck, Germany; 6Institute for Pathology and Molecular Pathology, German Armed Forces Hospital Ulm, Oberer Eselsberg 40, 89081 Ulm, Germany

**Keywords:** inflammation, risk scores, neuroendocrine neoplasms, GEP-NEN, CRP, albumin

## Abstract

**Simple Summary:**

There is growing evidence for the essential prognostic role of systemic inflammation within the tumor microenvironment (TME) and the nutritional status in cancer patients. Inflammation-based risk scores such as the Glasgow-Prognostic-Score (GPS), composed of C-reactive protein (CRP) and albumin levels at initial diagnosis, were shown to reflect the TME. This manuscript compares the prognostic impact of several well-established risk scores and ratios in the spectrum of neuroendocrine neoplasms of the gastro-entero-pancreatic (GEP-NEN) system. Our results highlight the prognostic capability of the GPS across the entire spectrum in GEP-NEN irrespective of histological grading or UICC stages and suggest its integration into more comprehensive models of risk stratification in the era of precision oncology.

**Abstract:**

Background: Across a variety of solid tumors, prognostic implications of nutritional and inflammation-based risk scores have been identified as a complementary resource of risk stratification. Methods: In this retrospective study, we performed a comparative analysis of several established risk scores and ratios, such as the Glasgow Prognostic Score (GPS), in neuroendocrine neoplasms of the gastro–entero–pancreatic (GEP-NEN) system with respect to their prognostic capabilities. Clinicopathological and treatment-related data for 102 GEP-NEN patients administered to the participating institutions between 2011 and 2021 were collected. Scores/ratios significantly associated with overall or progression-free survival (OS, PFS) upon univariate analysis were subsequently included in a Cox-proportional hazard model for the multivariate analysis. Results: The median age was 62 years (range 18–95 years) and the median follow-up period spanned 51 months. Pancreatic or intestinal localization at the initial diagnosis were present in 41 (40.2%) and 44 (43.1%) cases, respectively. In 17 patients (16.7%), the primary manifestation could not be ascertained (NNUP; neuroendocrine neoplasms of unknown primary). Histological grading (HG) revealed 24/102 (23.5%) NET/NEC (poorly differentiated; high grade G3) and 78/102 (76.5%) NET (highly or moderately differentiated; low–high grade G1–G2). In total, 53/102 (51.9%) patients presented with metastatic disease (UICC IV), 11/102 (10.7%) patients presented with multifocal disease, and 56/102 (54.9%) patients underwent a primary surgical or endoscopic approach, whereas 28 (27.5%) patients received systemic cytoreductive treatment. The univariate analysis revealed the GPS and PI (prognostic index), as well as UICC-stage IV, HG, and the Charlson comorbidity index (CCI) to predict both the PFS and OS in GEP-NEN patients. However, the calculation of the survival did not separate GPS subgroups at lower risk (GPS 0 versus GPS 1). Upon the subsequent multivariate analysis, GPS was the only independent predictor of both OS (*p* < 0.0001; HR = 3.459, 95% CI = 1.263–6.322) and PFS (*p* < 0.003; HR = 2.119, 95% CI = 0.944–4.265). Conclusion: In line with previous results for other entities, the present study revealed the GPS at baseline to be the only independent predictor of survival across all stages of GEP-NEN, and thus supports its clinical utility for risk stratification in this group of patients.

## 1. Introduction

At an incidence of 3.5:100,000 neuroendocrine neoplasms of the gastro–entero–pancreatic system (GEP-NEN) constitute a rare subgroup of solid, gastrointestinal tumors [1,2]. However, in recent years, the incidence of GEP-NEN has increased due to continued improvements in endoscopic, radiologic, and histopathologic diagnostics [3,4]. The heterogeneous group of GEP-NEN arise from cells related to the diffuse neuroendocrine system [5]. These cells exhibit typical features of endocrine and neuronal cells [6,7]. In the majority of cases, the proliferation of GEP-NEN cells is low to moderate [4,8]. The most frequent primary sites of GEP-NEN affect the pancreas (23%) and the gastrointestinal (GI) tract, including the stomach (6.5%), the jejunum (2.5%), ileum (21.0%), the appendix (6.8%), the colon (12.1%), and rectum (19.1%) [3,9]. Because of the slow and masked progression of the disease, loco-regional or hepatic metastases can be detected in up to 50% of cases at the initial diagnosis [4]. However, metastatic disease is more common in neuroendocrine carcinoma (NEC) of the gastro–entero–pancreatic system (GEP-NEC) rather than in GEP-NET. In pancreatic NEN, the rate of primary metastatic disease is 71.9% [10]. The functional activity in GEP-NEN is of clinical relevance in a notable subset of cases. Carcinoid syndrome, presenting with flush, diarrhea, abdominal cramps, and carcinoid-related heart disease (right-accentuated myocardial fibrosis), reflects the most frequent functional GEP-NEN manifestation [4,7,11]. The underlying pathophysiological mechanism is based on the secretion of serotonin (GI-symptoms), histamine, and/or bradykinin (flush) originating from GEP-NEN cells and avoiding hepatic inactivation [4].

Treatment guidance is based on histological grading and staging results [7,12]. Histological grading (grade 1–3) is determined by the current version of the World Health Organization (WHO) classification of digestive system tumors and factors in the Ki-67 proliferation index or the mitotic count, alongside the morphological degree of cellular differentiation (neuroendocrine tumor/NET (G1–G3) versus neuroendocrine carcinoma/NEC (G3)) [13]. Staging results are classified using the ENETS (European Neuroendocrine Tumor Society) TNM classification system and subsequently translated into the current UICC (Union for International Cancer Control) staging system [14,15].

The spectrum of therapeutic options is exhaustive. In localized stages, endoscopic or surgical resection displays the only therapeutic approach with a curative intent [7,12]. Surgical approaches also play a role in terms of disease control if tumors are functionally active or if they cause disruptive secondary organ infiltration [7]. Moreover, surgical resection should be considered in G1–G2 tumors presenting with hepatic metastasis [7]. Response rates to systemic chemotherapy in advanced stage (UICC IV) G3 NEN/NEC remain poor [16,17,18]. However, a 5-fluorouracil or platin-based chemotherapeutic regimen reflects an option with palliative intent [12,18]. In some subgroups, targeted therapeutics such as mTOR-inhibitors (everolimus) or tyrosine-kinase inhibitors (sunitinib or surufatinib) have demonstrated a promising efficacy [19,20,21,22,23]. Nuclear medical approaches such as peptide receptor radionuclide therapy (PRRT) or I-131-MIBG therapy, as well as interventional radiologic approaches (transarterial chemoembolization/TACE or radioembolization/selective internal radiation therapy (RE/SIRT)) and other pharmacological options containing somatostatin analogues (SSA; e.g., octreotide or lanreotide), interferon-alpha or serotonin synthesis inhibitors (e.g., telotristat) complete the field of treatment options [12,24,25,26,27,28]. In the case of unresectable, disseminated liver metastasis driving the prognosis in a NEN patient, selective internal radiation therapy (SIRT) can be preferred to TACE. For selected cases with diffuse metastasis of the liver and/or unsuccessful pharmacological efforts in terms of the control of functionally active tumors, liver transplantation reflects an ultimate treatment option [12]. However, treatment guidance is challenging and the preferable therapeutic approach should be determined in an interdisciplinary context.

As the optimal treatment guidance displays such a challenging process, it is of major importance to reliably identify GEP-NEN patients at risk. Several potential risk factors predicting adverse outcomes in GEP-NEN patients have been reported in the literature so far. These include the performance status at the initial diagnosis, advanced age, and elevated serum levels of the lactate dehydrogenase (LDH), as well as platelets, primary sites, and tumor size [29]. The cut-off values of Ki-67 for optimal risk stratification are discussed conversely [29,30,31]. Among a large spectrum of solid tumors and hematological malignancies, it has been shown that the Glasgow Prognostic Score (GPS) reflects a representative and handy tool to identify cancer patients at risk for both early progression and all-cause mortality [32,33,34,35,36,37,38,39,40,41,42]. GPS examines two essential mechanisms of tumor growth, progression and aggressiveness. The first is systemic inflammation (via the C-reactive protein) and the second is the nutritional status of a cancer patient (via albumin) at the initial diagnosis [35]. In recent years, the impressive predictive capabilities of both factors have been demonstrated in cancer patients. A recent study demonstrated the prognostic impact of a modified GPS in high grade GEP-NEN (G3) [43]. Unfortunately, the majority of studies investigating the predictors of survival in GEP-NEN exclusively included high-grade tumors. Here, we assessed the prognostic capabilities of GPS within the spectrum of several well established and validated nutritional- and/or inflammation-based risk scores or ratios in terms of optimal risk stratification across all stages and grades of GEP-NEN.

## 2. Methods

This retrospective multicenter study investigated the prognostic capabilities of several established and validated risk scores/ratios in GEP-NEN patients at the initial diagnosis. We screened all GEP-NEN patients that underwent surgical/endoscopic and/or cytoreductive treatment in one of the participating institutions (University Hospital Schleswig-Holstein (UKSH) Campus Lübeck and German Armed Forces Hospital Ulm) between 2011 and 2021. Initial screening in our institutional hospital information system (HIS) identified 144 patients diagnosed with GEP-NEN. Patients with insufficient follow-up (17 patients referred to other institutions within 30 days after initial diagnosis and subsequent loss of follow-up in 25 cases) were excluded. Moreover, paragangliomas and phaeochromocytomas were excluded as well. Staging was carried out in accordance with current ‘Arbeitsgemeinschaft der Wissenschaftlichen Medizinischen Fachgesellschaften e.V.’ (AWMF) guidelines [7].

### 2.1. Baseline Clinicopathological Characteristics

Clinical information was collected from the original electronic patient files. Besides sex and age, data collection included the evaluation of the Eastern Cooperative Oncology Group Performance Status (ECOG-PS), staging data with primary localization and localization of metastases where available, results from histopathological assessments including morphological pattern and immunohistochemical staining, laboratory data from initial diagnosis (see Section 2.2), and the further course of the disease, the presence of carcinoid syndrome, treatment modalities (first, second, and third line where available), and responses, as well as the pattern of relapse and survival data.

### 2.2. Prognostic Risk Scores/Ratios

The scores/ratios considered for the present study contained inflammatory and/or nutritional features such as the serological levels of different immune cells such as neutrophils and lymphocytes. Therefore, laboratory data incorporated parameters from the baseline differential blood count, the inflammation related parameter serum levels of albumin (g/dL), and the C-reactive protein (CRP; mg/dL), as well as the serological tumor marker chromogranin A. The composition of all of the scores and ratios is outlined in Table 1. Our calculations included the widely accepted neutrophil–lymphocyte ratio (NLR) and the platelet–lymphocyte ratio (PLR) [44,45,46,47]. Related scores that have been calculated were the neutrophil–lymphocyte score (NLS), the platelet–lymphocyte score (PLS), and the neutrophil–platelet score (NPS) [48]. Furthermore, the prognostic nutritional index (PNI) takes into account the patient’s nutritional status in terms of the serum albumin (PNI = albumin + 0.005 × total lymphocyte count) [36]. The prognostic index (PI) incorporates the white blood cell count (>10 × 10^9^/L) and the CRP (>10 mg/dL) reflecting the acute phase [49]. Considering both the acute phase and the nutritional aspect, for GPS calculation, a CRP > 10 mg/dL and/or an albumin value of <35 g/L counted as one point, resulting in three different subgroups (group I: 0 points; group II: 1 point; group III: 2 points) [33]. We also performed the calculation of the CRP–albumin ratio (CAR) [50].

### 2.3. Treatment and Responses

Stage-adapted treatment decisions were made on the basis of the interdisciplinary tumor board consensus for neuroendocrine neoplasms at UKSH Campus Lübeck (ENETS CoE) with current ENETS guidelines serving as an institutional standard [7]. Treatment response was defined in accordance with the established Response Evaluation Criteria in Solid Tumors (RECIST) v1.1 [51]. Progression-free and overall survival (PFS and OS) were calculated from the date of the initial diagnosis. Toxicity profiles were conducted in accordance with the National Cancer Institute Common Toxicity Criteria (NCI CTC; version 2.0) [52].

### 2.4. Ethics Statement

This retrospective study was approved by the ethics committee of the University of Luebeck (reference number 18-041) and was conducted in accordance with the Declaration of Helsinki. Written informed consent referred to routine diagnostics and academic assessment of the biopsy specimen, and the transfer of clinical data was obtained from all patients.

### 2.5. Statistics

All of the statistical analyses were conducted using GraphPad PRISM 9 (GraphPad Software Inc., San Diego, CA, USA), SPSS 26 (IBM, Armonk, NY, USA), and R v4.0.2. The Kolmogorov–Smirnov test was performed to assess the normality of distribution. Survival (PFS and OS) was estimated using the Kaplan–Meier method and the univariate log-rank test. A confirmatory univariate Cox analysis was subsequently performed. A subsequent multivariate proportional hazard model (Cox proportional hazard) was conducted for characteristics exhibiting a trend towards statistical significance (*p* < 0.07) that were found to be associated with OS or PFS upon both univariate analyses. However, the significance level was defined at *p* < 0.05. Cut-off values for NLR, CAR, and PNI were selected from previously published data investigating the prognostic impact of such scores across a variety of cancer patients. Moreover, cut-off value confirmation was performed utilizing a receiver operating characteristic (ROC) analysis proposed by Budczies et al. [53]. The Mann–Whitney U test and the Chi-squared test were used to assess differences between GEP-NEN subgroups, as appropriate. Pearson’s correlation was calculated in order to anticipate the non-linear relations between variables. Comparative analysis regarding the prognostic impact of the GPS and the CAR was performed by estimating the Harrel’s concordance index (c-index) and the corrected Akaike’s information criterion (cAIC) in order to identify the most qualified risk score considering both the acute phase and the nutritional aspect [54,55].

## 3. Results

### 3.1. Clinicopathological Characteristics

The clinicopathological features of the study group are outlined in Table 2. The median age at initial diagnosis was 62 years (range 18–95) and gender distribution within the study cohort was balanced (male 54.9%)/female 45.1%). Because of the heterogeneity of the study population as well as short follow-up periods in a relevant subset of patients without any relapse or death event (n = 22; 21.6%), the median follow-up was 25.0 months (1–165 months) and the median body mass index (BMI) was 26.89 kg/m^2^ (range 16.96–40.79 kg/m^2^). Moreover, 29 (40.8%; information was available in 71 patients) patients had significant weight disorders (cachexia (BMI < 20 kg/m^2^) = 9; obesity (BMI > 30 kg/m^2^) = 20). Decrease in medical fitness was present in 20 patients (19.6%) (elevated ECOG-PS ≥ 2). The pancreas (n = 41; 40.2%) and the intestine (n = 44; 43.1%; please see Table 2 for specific primary tumor sites) were the most frequent primary tumors in the present cohort. Further, 17 patients (16.7%) had a neuroendocrine neoplasm of an unknown primary (NNUP). Among the entire study cohort, 20 patients with GEP-NEC were included. Staging revealed metastatic disease in 53 patients (51.9%), most of whom had liver metastases. Multilocal disease was present in 11 cases (10.8%) at the initial diagnosis. The GPS was positively associated with histological grading (Table 2). The minority of patients experienced carcinoid syndrome (18.6%) or B-symptoms (24.5%). The median Ki-67 was 10.0%. In GEP-NEC, the median Ki-67 was 80% and in GEP-NET it was 5%. Additionally, the Ki-67 was positively correlated with histological grading and GPS. Figure 1 visualizes the distribution of the relevant clinicopathological characteristics in the GPS subgroups.

### 3.2. Prognostic Scoring Systems

The distribution of the composite scores/ratios and their component values are depicted in Table 3. It appears that only the minority of GEP-NEN patients initially showed features of systemic inflammation (NLR > 5 (28.4%); NLS = 2 (16.7%); NPS = 2 (3.9%); PLS = 2 (0.98%); PLR > 150 (57.8%); PI = 2 (11.7%); PNI > 50 (33.3%); CAR ≥ 0.22 (34.3%); and GPS = 2 (16.7%)).

Pearson’s correlation analysis revealed the close connection between the GPS and increasing UICC stage and other inflammation and/or nutritional aspects reflecting scores/ratios such as the CAR, PI, NPS, PLS, NLS, NLR, and PLR. Moreover, correlations between the GPS and the occurrence of B-symptoms were detected. More results of Pearson’s correlation analysis are demonstrated in Figure 2.

The univariate Cox analysis revealed the GPS, PI, CCI > 3, metastatic disease (UICC IV), and high-grade histology (NEC G3) to potentially predict PFS and OS (Table 4). Upon univariate analysis, CRP and albumin as individual components of the GPS and CAR were found to significantly impact survival (CRP: *p* = 0.0016 (OS), *p* = 0.005 (PFS); albumin: *p* = 0.013 (OS)), which is in concurrence with previous results investigating the impact of nutritional- and inflammation-based risk scores in hematological malignancies.

A subsequent confirmatory multivariate analysis conducting a Cox proportional hazard model was conducted for the characteristics, scores, or ratios that significantly predicted either OS or PFS after the univariate analysis (Table 5). As the GPS and the CAR incorporated serological levels of CRP and albumin, the calculation of the c-index and the cAIC was performed to identify the superior CRP/albumin-based scoring system for multivariate analysis. This analysis demonstrated that the GPS outperformed the CAR (Appendix A).

The subsequent comparative multivariate analysis showed that the GPS (HR = 3.459, 95% CI = 1.263–6.322, *p* < 0.0001), ECOG-PS (HR = 1.667, 95% CI = 0.828–4.189, *p* = 0.042), metastatic disease (UICC IV) (HR = 1.155, 95% CI = 0.870–1.399, *p* = 0.001), and NEC histology (HR = 1.271, 95% CI = 0930–1.661, *p* = 0.002) had a significant influence on OS, whereas only the GPS (HR = 2.119, 95% CI = 0.944–4.265, *p* = 0.003) significantly predicted PFS. Consequently, GPS was identified as the exclusive independent predictor of both OS and PFS. Moreover, subsequent multivariate analysis could not confirm the significant dichotomized predictive value of the PI, PNI, CCI > 3, metastatic disease (UICC IV), and histological grading (NEC G3) from the univariate analysis.

The impact of GPS and CAR on OS (*p* = 0.003; *p* = 0.003) and PFS (*p* = 0.0085; *p* < 0.001) is demonstrated by the Kaplan–Meier analysis in Figure 3. The additional Kaplan–Meier analysis revealed that the histological grading had no impact on either PFS or OS within the GPS subgroups, with the exception of calculating the OS in the GPS 0 subgroup (*p* = 0.0009). Moreover, NNUP had a significant impact on OS (*p* = 0.0006) but not PFS (0.093) (Figure 3).

Appendix A shows that GPS subdivided high-risk patients (GPS 2) from low- and intermediate-risk (GPS 0/1) patients in pancreatic NEN (pan-NEN), but not in the NEN of the small intestine (SI-NEN) (Appendix A). The high frequency of G1 and G2 tumors (96.0%) in SI-NEN can explain the failure of risk prediction in this NEN subtype. As expected, the present analysis found the highest frequency of high-risk patients (GPS 2) in NECs (Appendix A).

During the follow-up period of 25.0 months in the median, the median GPS-subgroup-related PFS was 22 months (GPS 0), 14 months (GPS 1), and 13 months (GPS 2), respectively. In this period, 51 GEP-NEN patients experienced a relapse event (29.4%) or refractory setting (20.6%), and 39 cancer-related deaths (38.2%) were recorded. GEP-NEN patients with a calculated GPS of 2 had a higher relapse rate of 58.8% compared with the lower GPS subgroups (GPS 0 = 33.9%; GPS 1 = 42.3%). Finally, the calculated two-year OS rate was 57.6% for the GPS 0 subgroup, 40.7% for the GPS 1, and 43.7% for the GPS 2 subgroup, respectively.

### 3.3. Treatment Characteristics

The treatment modalities related to histological grading, associated response rates, and the toxicity profile in the current study cohort are outlined in Table 6. Surgical resections display the therapeutic approach that was performed the most frequently (63.7%). As Table 6 demonstrates, in 23 cases (22.5%), surgical approaches were conducted in a palliative setting and in 42 cases (41.2%) the approach was performed with curative intent. However, 25 GEP-NEN patients (24.5%) received palliative chemotherapy and the target therapeutic everolimus was administered in two cases (1.9%) in a first line setting. Of course, systemic treatment in palliative intent was administered the most frequently in patients with NEC (75%). The most frequent chemotherapeutic regimen was a platin-based protocol (64.0%). Somatostatin analogues were given in the majority of cases (n = 23; 22.5%). A PRRT was carried out in 13 cases (12.7%). Seven patients (6.9%) refused any cytoreductive treatment. Further treatment information for relapse settings are briefly summarized in Appendix A and in the Appendix A.

Considering the RESIST v1.1 criteria, the overall response rate (ORR) was 70.9% (CR and PR). The CR rate in this cohort was 38.7% (36/93) and the rate of PRs was 32.2%. (30/93). There were progressive diseases in 7.5% of cases (7/93) without any response to cytoreductive treatment. A watch and wait strategy was favored in eight cases (7.8%).

Although the majority of patients presented with advanced stage disease at initial diagnosis and experienced intensive treatment strategies, the overall toxicity profile was mild and mostly gastrointestinal in nature with emesis and/or nausea. Severe adverse events (SAE) were recorded in 14 cases (15.1%) in the first line setting.

## 4. Discussion

The present study comprehensively compares nutritional and inflammatory risk scores/ratios with respect to their prognostic capabilities in GEP-NEN patients. Especially in GEP-NEN, an extremely wide prognostic spectrum exists, depending not only on the stage at initial diagnosis, but also on the applicability of the exhaustive therapeutic repertoire in each individual case [29]. Accordingly, there is significant room for improvement when identifying GEP-NEN patients at risk of early progression in order to provide each patient with a suitable therapeutic strategy. Alongside a variety of several malignancies, the impact of GPS and its impressive role as a complementary resource for risk stratification has been demonstrated [33,36,38,50]. In the era of precision oncology and comprehensive genomic analysis, risk stratification of malignant diseases has enormously evolved in recent years. However, because of financial issues, comprehensive genomic analyses are not available for the vast majority of tumor entities to date. Additionally, molecular analysis is associated with a certain latency. Consecutively, simple but informative resources for risk stratification are required to identify cancer patients at risk at an early stage. Most scores/ratios that have been created for prognostic prediction in cancer solely consider the aspect of inflammation (NLR, NLS, NPS, PLR, PLS, and PI). Previous studies have investigated the role of inflammation-based risk scores/ratios in advanced or metastatic NEN, noting the prognostic role of increased CRP and white blood cell counts, reflecting an inflamed tumor microenvironment [56]. The authors did not find merit in the calculation of NLR and PLR for outcome prognostication. Adding the nutritional aspect to a prognostic score/ratio extends its prognostic capability, as several studies have shown for CAR, PNI, and GPS [57,58]. CAR and GPS can be calculated on the basis of routine laboratory parameters. Therefore, their calculation is a readily available and extremely cost-effective tool to predict survival in cancer patients. In their study, Dolan et al. highlighted the superiority of scores compared with ratios for risk stratification in cancer. However, CAR showed a promising predictive power with regard to PFS and OS [36,50]. To avoid the redundancy of individual components and the expected loss of independency of GPS upon further multivariate analysis, we initially compared both CRP and albumin-based risk scores (GPS and CAR) by calculating the c-index and the cAIC. In this context, we identified GPS to hold more accurate prognostic capabilities. This led to the exclusion of CAR from our subsequent multivariate analysis.

Upon comparative multivariate analysis, GPS outperformed the well-established risk scores/ratios and clinical characteristics underlining its prognostic independence for both PFS and OS. As previously shown, a modified GPS (mGPS) predicts prognosis in high-grade NEN of any primary site [43]. In the mGPS, hypoalbuminemia alone does not count as a risk factor. Just the combination with an elevated CRP defines a constellation of higher risk. Therefore, in the present cohort, mGPS was able to identify low-risk but not high-risk patients (Appendix A). Here, we focused on NEN from the gastro–entero–pancreatic system using data from molecular profiling, which highlighted distinct genomic programming for each primary NEN site and expanded the spectrum through all histological grades and UICC stages [59]. Our results did not confirm the prognostic capabilities of Ki-67 at the initial diagnosis that were proposed by Abdelmalak et al. [43]. The authors suggested that the prognostic ability of mGPS may be grade-dependent in NEN, as Zou et al. were not able to show the prognostic relevance of the mGPS among several histological grades in their cohort [43,56]. However, in our cohort, which is of comparable sample size (n = 102 versus n = 135), we evaluated the prognostic independence of the GPS across a wider spectrum of inflammation and nutritional status-based risk scores/ratios.

The Kaplan–Meier survival analysis revealed significant distinction of some GPS subgroups. The GPS did not adequately separate the OS survival curves of patients at low (GPS 0) and intermediate (GPS 1) risk, which is in agreement with previous results [35,43]. PFS analysis demonstrated a distinction for each GPS subgroup, underlining that both GPS components, inflammation and nutritional status, had a relevant impact on selective GEP-NEN prognosis, but not necessarily all-cause mortality. In comparison with the inflammation-based risk scores/ratios, these findings line up with the assumption of previous studies that the inclusion of a second dimension allows for a clearer distinction of risk groups across a variety of cancer subtypes [43]. Moreover, the relapse rate increased with a higher GPS score. As a consequence, the calculation of GPS in GEP-NEN patients can potentially influence treatment guidance in cases with several therapeutic options of distinct intensity, as such scores reflect a relevant component to identify cancer patients at risk.

The results from the molecular profiling revealed specific genomic signatures for pancreatic (*MEN1*, *ATRX,* and *CREBBP*), midgut (*CDKN1B*) NETs, and GEP-NEC (*TP53*, *RB1*, *KRAS*, *CSMD3*, *TRRAP*, and *MYC*) [59,60]. For optimal risk stratification, results from these more effortful diagnostics will be complementary to conclusions that can be drawn from GPS calculation.

The potential limitations of the present study include its limited sample size and its retrospective design harboring the perpetual eventuality of fragmentary data alongside heterogeneous treatment approaches. Because of the fragmentary data sets, information on neuron specific enolase (NSE) and the secretion of serotonin, gastrin, insulin, or glucagon was available only in a limited subset of patients, as the measurement of hormones at the initial diagnosis remains optional. Although we were able to evaluate the causes of death in the majority of cases, the cause of dead remained unknown for a subset of patients due to insufficient follow-up. Moreover, 22 patients with a short follow-up period (<24 months) were included for a short median-follow-up of not more than 25 months. Concurrent infections at initial diagnosis harbor the potential to distort the calculation of the GPS. Hence, GEP-NEN patients considered for inclusion in the study were screened for infections that possibly bias the scoring results. A period of 30 days after the initial clinical presentation was acknowledged so as to determine an alternative date for another blood sampling in order to exclude any relevant infection affecting the calculation of the GPS.

## 5. Conclusions

Present results confirm the robustness of GPS as an excellent contributor for individual risk stratification in GEP-NEN patients independent from histologic grading. Adding clinical insights as well as specific features from the molecular characterization of pancreatic and midgut NETs and GEP-NEC to concepts of risk stratification in order to optimize the personalized prediction of prognosis in the era of precision oncology should be evaluated in further studies. To the best of our knowledge, this is the first study that underlines the prognostic capability of GPS among the entire spectrum of histological grading and UICC stages in GEP-NEN/NEC. However, further studies are required to validate the present results in a prospective setting including a larger sample size.

## Figures and Tables

**Figure 1 cancers-14-05465-f001:**
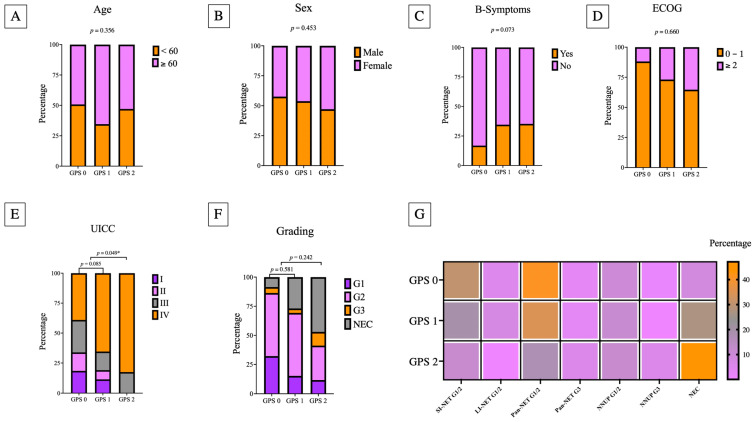
Stacked bar plots demonstrate the distribution of the relevant clinicopathological insights for each GPS subgroup ((**A**) age; (**B**) sex; (**C**) B symptoms; (**D**) ECOG performance status; (**E**) UICC stage; (**F**) histological grading). (**G**) The distribution of the GPS in relation to the primary tumor sites. Significant correlations are marked with * (*p* < 0.05).

**Figure 2 cancers-14-05465-f002:**
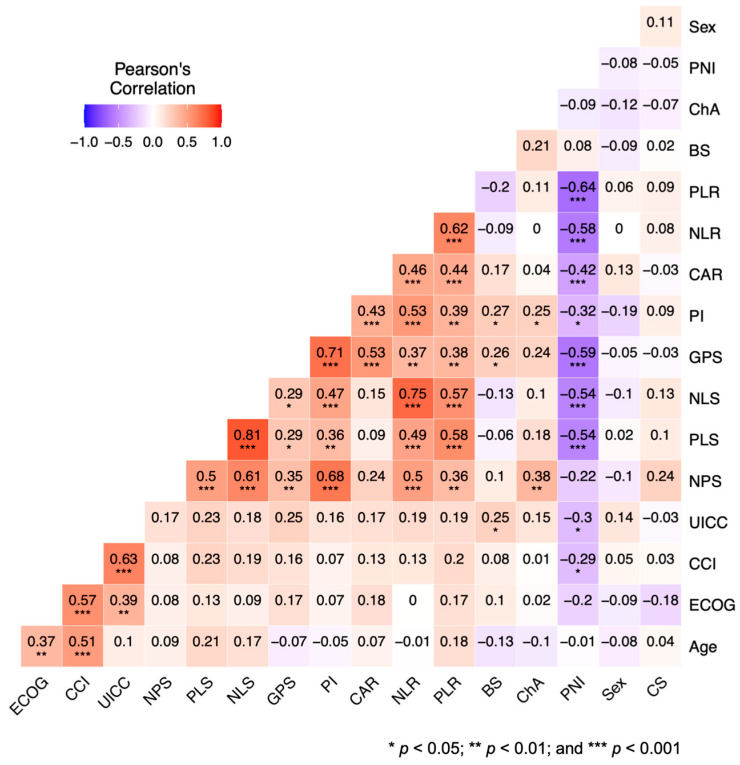
Visualization of Pearson’s correlation with respect to the clinical characteristics and inflammation-/nutritional-status-based risk scores/ratios. High degrees of correlation are colored in red and low degrees of correlation are colored in blue.

**Figure 3 cancers-14-05465-f003:**
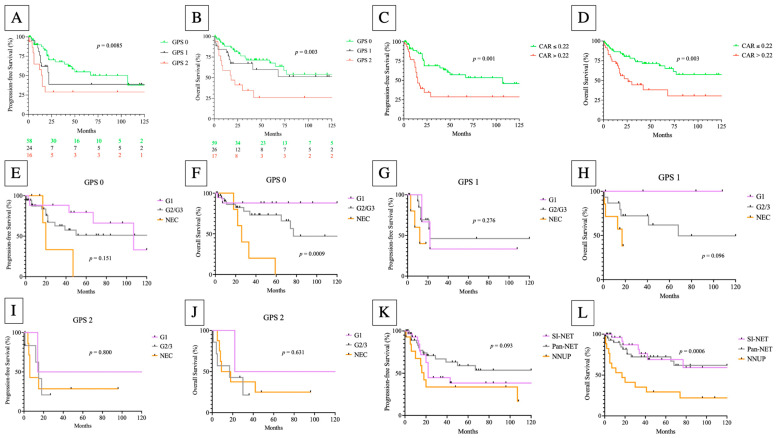
Progression-free (**A**,**C**,**E**,**G**,**I**,**K**) and overall (**B**,**D**,**F**,**H**,**J**,**L**) survival according to the Glasgow Prognostic Score (GPS) (log-rank GPS 0 vs. GPS 1 vs. GPS 2; (**A**,**B**)), CRP/albumin ratio (CAR) (log-rank test; (**C**,**D**)) and primary tumor sites (log-rank test; (**K**,**L**)) in GEP-NEN patients. (**E**–**J**) The Kaplan–Meier analysis (PFS and OS) according to the histological grading among the GPS subtypes.

**Table 1 cancers-14-05465-t001:** Systemic-inflammation-based prognostic ratios and scores.

Ratio/Score	Ratio/Score
NLR	
Neutrophil count:lymphocyte count	≤3
Neutrophil count:lymphocyte count	3–5
Neutrophil count:lymphocyte count	>5
NLS	
Neutrophil count ≤ 7.5 × 10^9^/L and lymphocyte count ≥ 1.5 × 10^9^/L	0
Neutrophil count > 7.5 × 10^9^/L and lymphocyte count ≥ 1.5 × 10^9^/L	1
Neutrophil count ≤ 7.5 × 10^9^/L and lymphocyte count < 1.5 × 10^9^/L	1
Neutrophil count > 7.5 × 10^9^/L and lymphocyte count < 1.5 × 10^9^/L	2
PLR	
Platelet count:lymphocyte count	≤150
Platelet count:lymphocyte count	>150
PLS	
Platelet count ≤ 400 × 10^9^/L and lymphocyte count ≥ 1.5 × 10^9^/L	0
Platelet count > 400 × 10^9^/L and lymphocyte count ≥ 1.5 × 10^9^/L	1
Platelet count ≤ 400 × 10^9^/L and lymphocyte count < 1.5 × 10^9^/L	1
Platelet count > 400 × 10^9^/L and lymphocyte count < 1.5 × 10^9^/L	2
PI	
White blood cell count ≤ 10 × 10^9^/L and C-reactive protein ≤ 10 mg/L	0
White blood cell count ≤ 10 × 10^9^/L and C-reactive protein > 10 mg/L	1
White blood cell count > 10 × 10^9^/L and C-reactive protein ≤ 10 mg/L	1
White blood cell count > 10 × 10^9^/L and C-reactive protein > 10 mg/L	2
PNI	
Albumin (g/L) + 5 × (lymphocyte count (10^9^/L))	≤50
Albumin (g/L) + 5 × (lymphocyte count (10^9^/L))	>50
NPS	
Neutrophil count ≤ 7.5 × 10^9^/L and platelet count < 400 × 10^9^/L	0
Neutrophil count > 7.5 × 10^9^/L and platelet count < 400 × 10^9^/L	1
Neutrophil count ≤ 7.5 × 10^9^/L and platelet count > 400 × 10^9^/L	1
Neutrophil count > 7.5 × 10^9^/L and platelet count > 400 × 10^9^/L	2
CAR	
C-reactive protein:albumin	≤0.22
C-reactive protein:albumin	>0.22
GPS	
C-reactive protein ≤ 10 mg/L and albumin ≥ 35 g/L	0
C-reactive protein > 10 mg/L or albumin < 35 g/L	1
C-reactive protein > 10 mg/L and albumin < 35 g/L	2

NLR, neutrophil–lymphocyte ratio; NLS, neutrophil–lymphocyte score; CAR, C-reactive protein albumin ratio; GPS, Glasgow Prognostic Score; NPS, neutrophil–platelet score; PLR, platelet–lymphocyte ratio; PLS, platelet–lymphocyte score.

**Table 2 cancers-14-05465-t002:** Baseline clinicopathological characteristics in the current study cohort.

GPS	Overall Study Group (n = 102)	Group I GPS 0 (n = 59)	Group II GPS 1 (n = 26)	Group III GPS 2 (n = 17)
Male/female	56/46	34/25	14/12	8/9
Median age (range), years	62 (18–95)	59 (18–85)	65 (36–95)	60 (23–81)
BMI (median, range)	26.9 (16.9–40.8)	28.0 (16.9–38.0)	26.0 (18.0–40.8)	26.0 (18.0–37.1)
Weight disorder				
Cachexia (BMI < 20 kg/m^2^)	9/71 (12.7%)	4/39 (10.3%)	4/19 (21.1%)	1/10 (10.0%)
Obesity (BMI > 30 kg/m^2^)	20/71 (28.2%)	14/39 (35.9%)	3/19 (15.8%)	3/10 (30.0%)
ECOG PS				
0–1	82 (80.4%)	52 (88.1%)	19 (73.1%)	11 (64.7%)
2–4	20 (19.6%)	7 (11.9%)	7 (26.9%)	6 (35.3%)
CCI (Median, range)	6 (0–13)	5 (0–12)	7.5 (2–11)	7 (4–13)
B-symptoms				
No	77 (75.5%)	49 (83.1%)	17 (65.4%)	11 (64.7%)
Yes	25 (24.5%)	10 (16.9%)	9 (34.6%)	6 (35.3%)
Primary sites				
Pancreatic	41 (40.2%)	27 (45.8%)	9 (34.6%)	5 (29.4%)
Intestine	44 (43.1%)	24 (40.7%)	13 (50.0%)	7 (41.2%)
- Gastric	7 (6.9%)	4 (6.8%)	1 (3.8%)	2 (11.8%)
- Jejunoileal	25 (24.5%)	15 (25.4%)	7 (26.9%)	3 (17.6%)
- Appendix	5 (4.9%)	4 (6.8%)	1 (3.8%)	-
- Colon	3 (2.9%)	1 (1.7%)	1 (3.8%)	1 (5.9%)
- Rectum	4 (3.9%)	-	3 (11.5%)	1 (5.9%)
Unknown Primary	17 (16.7%)	8 (13.5%)	4 (15.4%)	5 (29.4%)
Multifocal	11 (10.8%)	6 (10.1%)	3 (11.5%)	2 (11.7%)
Metastasis				
Yes	53 (51.9%)	23 (39.0%)	17 (65.4%)	13 (76.5%)
Carcinoid Syndrome				
No	83 (81.4%)	47 (79.7%)	20 (76.9%)	16 (94.1%)
Yes	19 (18.6%)	12 (20.3%)	6 (23.1%)	1 (5.9%)
Albumin (g/L) (median, range)				
≥35 g/L	71 (69.6%)	55 (93.2%)	15 (57.7%)	1 (5.9%)
<35 g/L	31 (30.4%)	4 (6.8%)	11 (42.3%)	16 (94.1%)
CRP (mg/dL) (median, range)				
≤10 mg/dL	72 (70.6%)	58 (98.3%)	13 (50.0%)	1 (5.9%)
>10 mg/dL	30 (29.4%)	1 (1.7%)	13 (50.0%)	16 (94.1%)
Chromogranin A median (range)	179 (29–56,200)	155 (29–13,600)	209 (41.2–8856)	196 (45–56,200)
Histological Grading				
NET (G1)	24 (24.7%)	18 (32.7%)	4 (16.0%)	2 (11.8%)
NET (G2)	49 (50.5%)	30 (54.5%)	14 (56.0%)	5 (29.4%)
NET(G3)	6 (5.9%)	3 (5.1%)	1 (3.8)	2 (11.8%)
Ki-67 (median, range)	5% (1–40%)	4% (1–30%)	5% (1–20%)	5% (1–40%)
NEC	20 (19.6%)	5 (8.5%)	7 (26.9%)	8 (47.1%)
- Small cell type	17 (16.7%)	5 (8.5%)	5 (19.2%)	7 (41.2%)
- Large cell type	3 (2.9%)	-	2 (7.7%)	1 (5.9%)
Ki-67 (median, range)	80% (40–90%)	80% (60–80%)	80% (40–80%)	75% (40–90%)
SSTR2				
Negative	30 (41.1%)	15 (34.9%)	8 (40.0%)	7 (70.0%)
Positive	43 (58.9%)	28 (65.1%)	12 (60.0%)	3 (30.0%)
UICC				
I	14 (13.7%)	11 (18.6%)	3 (11.5%)	-
II	11 (10.8%)	9 (15.3%)	2 (7.7%)	-
III	24 (23.5%)	16 (27.1%)	5 (19.2%)	3 (17.6%)
IV	53 (51.9%)	23 (39.0%)	16 (61.5%)	14 (82.4%)

BMI, body mass index; CCI, Charlson Comorbidity Index; CRP, C-reactive protein; ECOG PS, Eastern Cooperative Oncology Group performance status; GPS, Glasgow Prognostic Score; MEN, multiple endocrine neoplasia; NEC, neuroendocrine carcinoma; NET, neuroendocrine tumor; NSE; neuronspecific enolase; SSTR2, somatostatin-receptor-subtype 2; UICC, Union internationale contre le cancer.

**Table 3 cancers-14-05465-t003:** The relationship between composite ratios and cumulative scores and their component values in GEP-NEN (n = 102).

	n (%)	Median (Range)	Median (Range)
NLR		Neutrophils (×10^9^/L)	Lymphocytes (×10^9^/L)
<3	42 (41.6%)	4.1 (2.4–6.6)	2.1 (1.3–6.2)
3–5	30 (29.7%)	4.9 (2.9–10.2)	1.4 (0.6–2.9)
>5	29 (28.7%)	8.7 (4.9–19.7)	0.9 (0.4–2.7)
NLS			
0	39 (38.6%)	4.8 (2.4–7.4)	2.1 (1.5–6.2)
1	45 (44.6%)	4.9 (2.5–15.1)	1.3 (0.5–2.9)
2	17 (16.8%)	9.5 (7.9–19.7)	0.8 (0.4–1.4)
NPS		Neutrophils (×10^9^/L)	Platelets (×10^9^/L)
0	74 (73.3%)	4.5 (2.4–7.4)	250 (127–396)
1	23 (22.7%)	9.0 (4.1–19.7)	277 (137–595)
2	4 (3.9%)	9.4 (7.6–15.1)	425 (406–555)
PLR		Platelets (×10^9^/L)	Lymphocytes (×10^9^/L)
≤150	42 (41.6%)	233 (124–406)	2.1 (0.9–6.2)
>150	59 (58.4%)	267 (137–595)	1.2 (0.4–2.9)
PLS			
0	40 39.6%)	267 (168–378)	2.1 (1.5–6.2)
1	60 (59.4%)	250 (127–595)	1.2 (0.4–2.9)
2	1 (0.9%)	-	-
CAR		Albumin (g/L)	CRP (mg/dL)
≤0.22	67 (65.7%)	40 (30–51)	2.0 (0.0–9.4)
>0.22	35 (34.3%)	35 (18–46)	26.5 (6.0–242.0)
GPS			
0	59 57.8%)	40 (31–51)	2.0 (0.0–8.1)
1	26 (25.5%)	36 (25–46)	9.9 (0.3–98.4)
2	17 (16.7%)	31 (18–35)	40.7 (10.0–242.0)
PNI		Albumin (g/L)	Lymphocytes (×10^9^/L)
≥50	34 (33.7%)	45 (31–51)	1.9 (0.8–6.2)
<50	67 (66.3%)	36 (18–44)	1.2 (0.4–2.9)
PI		WBC (×10^9^/L)	CRP (mg/dL)
0	63 (61.7%)	7.0 (4.1–10.8)	2.2 (0.0–40.7)
1	27 (26.5%)	9.9 (4.8–17.8)	11.7 (0.6–186.0)
2	12 (11.7%)	12.5 (10.1–22.3)	36.6 (12.6–242.0)

NLR, neutrophil–lymphocyte ratio; NLS, neutrophil–lymphocyte score; NPS, neutrophil–platelet-score; PLR, platelet–lymphocyte ratio; PLS, platelet–lymphocyte score; CAR C, reactive protein albumin ratio; CRP, C-reactive protein; GPS, Glasgow Prognostic Score; PI, prognostic index; PNI, prognostic nutritional index; WBC, white blood cell count.

**Table 4 cancers-14-05465-t004:** Progression-free and overall survival in the univariate analysis (Univariate cox analysis).

Univariate Analysis				
	PFS		OS	
Prognostic Factor	*p*-Value	HR (95% CI)	*p*-Value	HR (95% CI)
GPS	**<0.0001**	4.479 (2.302–8.716)	**<0.0001**	6.153 (3.181–11.90)
CRP	**0.005**	1.009 (1.003–1.016)	**0.016**	1.008 (1.001–1.014)
Albumin	0.08	0.950 (0.898–1.006)	**0.013**	0.931 (0.881–0.985)
NLR	0.493	1.016 (0.972–1.061)	0.096	1.032 (0.994–1.071)
PLR	0.741	0.999 (0.007–1.002)	0.268	1.001 (0.991–1.003)
PNI	0.065	0.963 (0.924–1.002)	**0.005**	0.942 (0.904–0.982)
PI	**0.02**	1.675 (1.093–2.566)	**0.02**	1.663 (1.083–2.552)
Age > 60 years	0.237	1.455 (0.782–2.707)	0.078	1.793 (0.938–3.430)
ECOG PS ≥ 2	0.08	1.860 (0.911–3.800)	**<0.0001**	3.668 (1.935–6.950)
CCI > 3	**0.004**	4.608 (1.638–12.97)	**0.006**	7.418 (1.786–30.81)
UICC IV	**<0.0001**	1.698 (0.903–3.194)	**<0.0001**	1.292 (0.689–2.424)
NEC (G3)	**0.0004**	3.168 (1.161–8.646)	**<0.0001**	3.817 (1.548–9.412)

CCI, Charlson Comorbidity Index; CRP, C-reactive protein; ECOG PS, Eastern Cooperative Oncology Group performance status; GPS, Glasgow Prognostic Score; HR, hazard ratio; NLR, neutrophil–ratio; OS, overall survival; PFS, progression-free survival; PI, prognostic index; PLR, platelet–lymphocyte ratio; PNI, prognostic nutritional index; UICC, Union International Contre le Cancer. Bold values indicate statistical significance (*p* < 0·05) in the univariate cox analysis.

**Table 5 cancers-14-05465-t005:** Overall survival and progression-free survival in the univariate analysis and consecutive multivariate Cox proportional hazard regression.

	Univariate Analysis OS	Multivariate Analysis OS	
Prognostic Factor	*p*-Value	*p*-Value	HR (95% CI)
GPS	**<0.0001**	**<0.0001**	3.459 (1.263–6.322)
PI *	**0.02**	0.690	2.344 (1.513–8.436)
PNI **	**0.005**	0.409	0.851(0.535–4.331)
ECOG	**<0.0001**	**0.042**	1.667 (0.828–4.189)
CCI > 3	**0.006**	0.530	0.715 (0.299–6.299)
UICC IV	**<0.0001**	**0.001**	1.155 (0.870–1.399)
NEC (G3)	**<0.0001**	**0.004**	1.271 (0.930–1.661)
	**Univariate Analysis PFS**	**Multivariate Analysis PFS**	
	***p*-Value**	***p*-Value**	**HR (95% CI)**
GPS	**<0.0001**	**0.002**	2.119 (0.944–4.265)
PI *	0.02	0.518	2.775 (1.984–4.372)
PNI **	0.065	0.644	1.384 (1.015–1.855)
ECOG	0.08	0.453	2.248 (1.433–3.556)
CCI > 3	**0.004**	0.092	4.210 (1.936–6.501)
UICC IV	**<0.0001**	0.067	1.582 (1.214–2.737)
NEC (G3)	**0.0004**	0.081	1.322 (0.862–2.453)

CCI, Charlson Comorbidity Index; GPS, Glasgow Prognostic Score; OS, overall survival; PFS, progression-free survival; PI, prognostic index; PNI, prognostic nutritional Index; UICC, Union InterNational Contre le Cancer. * CRP > 10 mg/dL, white blood cell count > 11,000/μL, ** >50.

**Table 6 cancers-14-05465-t006:** First line treatment modalities of all GEP-NEN-patients included in the study.

Characteristics	Overall Study Group (n = 102)	GPS 0 (n = 59)	GPS 1 (n = 26)	GPS 2 (n = 17)
G1–G2 GEP-NEN 1st line treatment (n = 76)				
Surgical resection	52	38	11	3
- curative	37	30	5	2
- palliative	15	8	6	1
Chemotherapy	8	5	-	3
Targeted therapy	-	-	-	-
Radiation therapy	1	-	1	-
PRRT	12	7	5	-
Somatostatin analogues	22	11	8	3
Refusal	4	1	1	2
G3 GEP-NEN 1st line treatment (n = 6)				
Surgical resection	1	-	1	-
- curative	1	-	1	-
Chemotherapy	3	1	-	2
Targeted therapy	1	1	-	-
PRRT	1	1	-	-
Somatostatin analogues	2	1	-	1
GEP-NEC 1st line treatment (n = 20)				
Surgical resection	12	3	5	4
- curative	4	-	3	1
- palliative	8	3	2	3
Chemotherapy	14	5	6	3
Targeted therapy	1	1	-	-
Radiation therapy	1	-	-	1
Refusal	3	-	2	1
Best response (RECIST v1.1)				
CR	36	28	6	2
PR	30	11	10	9
SD	19	11	6	2
PD	7	2	4	1
Watch & wait	10	7	-	3
Dfd	39	17	10	12
Toxicity profile (NCI CTC)				
Cytopenia grad III/IV	5	2	2	1
Emesis	5	-	3	2
Pneumonitis	1	1	-	-
Nephrotoxicity	2	-	-	1

CR, complete remission; Dfd, death from disease; GPS, Glasgow-prognostic score; NCI CTC, National Cancer Institute Common Toxicity Criteria; PD, progressive disease; PR, partial remission; PRRT, peptide-receptor-radionuclide-therapy; SD, stable disease; RECIST, response evaluation criteria in solid tumors.

## Data Availability

The data that support the findings of this study are available from the corresponding author (H.M.W.), upon reasonable request.

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
