# Peer review of "The Glasgow Prognostic Score Predicts Survival Outcomes in Neuroendocrine Neoplasms of the Gastro–Entero–Pancreatic (GEP-NEN) System"

_cancers, 2022, doi:10.3390/cancers14215465_

Round 1

Reviewer 1 Report

This paper by Gebauer and colleagues analyzed the Glasgow-Prognostic-Score (GPS) (among other scores) in GEP-NEN and found it to be predictive of survival outcomes.

Summary:

The findings of the study are generally interesting. However I have some  concerns that need to be adressed before the paper can be published. My major concern is that GEP-NEN represent a wide spectrum of fundamentally different neoplasms, which differ between site (small intestine vs pancreatic) and also between NET and NEC. This is not properly addressed in the current form of the manuscript. The authors should reavaluate their findings with a more accurate consideration of tumor histology and should also perform a separate, site specific analysis in pancreatic and intestinal NEN (uni- and multivariate, detailed comments see below).

Major comments:

1.Regarding the histology, it is not correct to summarize NETG3 and NEC into one group, as NET G3 are not poorly differentiated neoplasms. Although they show an accentuated proliferation, they show an organoid, well differentiated growth pattern and are associated with a much better prognosis than NEC as they are genetically essentially different from them (see literature from pancreas for example!). I suggest regrouping into NETG1/2, NETG3 and NEC (also small cell vs. large cell if this information is available).

This should also be incorporated into all statistical analyses including the multivariate models!

2.Please further specify the localization of the investigated NEN. Intestine=jejunoileal? Were multifocal NET present?

3. it has to be noted that a median follow-up of 25 months is relatively short. Please explain.

4. Are there any differences between GPS in NETG1/2, NETG3 and NEC?

5. Are there differences of GPS score between localizations? Accentuated in CUP-NEN? Please provide P-values for Figure 1.

6. I would also suggest a separate prognostic analysis for pancreatic and intestinal NEN. SI-NET are usually low-grade NET with a very long survival while pNEN show a more agressive course and have a significantly higher rage of NETG3 and also NEC.

7. In the Discussion it is stated that TP53/RB1/CKN2A drive GEP-NEN, this is not precise, these are drivers of NEC and are of prognostic relevance as they are usually not found in NET. Some NET can obtain p53 alterations during disease progession (in Transit NET), but it is very unusual, even in NET G3 (see literature, very little to no mutational overlap between NET and NEC).

Author Response

We are grateful for the constructive evaluation of the present manuscript. Through this revision, we now hope to have provided all necessary amendments and clarifications.

  1. We totally agree with the reviewer’s concerns regarding the clear separation of neuroendocrine carcinomas (NEC) and neuroendocrine tumors (G3). Following the reviewer’s instructions, we now separate the 20 NECs from the six cases of G3 NETs included in the present study (page 5, line 207; lines 211 - 212; Table 2; Figure 1; Figure 2). Moreover, we provide information regarding small cell and large cell types in NEC (Table 2). As a consequence, we performed additional statistical analyses to adequately address the issue raised here (page 9, lines 241 - 242; Table 4 and Table 5; page 11, lines 268 – 271; Figure 3; Supplementary Figure 1).
  2. Of course, we added further specifications of primary tumor localizations and information regarding multifocal NET manifestations (page 5, lines 208 – 209 and Table 2).
  3. We thank the reviewer for calling attention to the limited follow-up period of our GEP-NEN/NEC cohort. We added a section discussing the concerns regarding this limitation in the discussion section (page 5, lines 197 – 199; page 14, lines 392 -393).
  4. To address the issue raised by the reviewer, we now provide additional analyses as outlined in Figures 1 and 3 as well as Supplementary Figure 1 E (page 11, lines 281 – 284).
  5. As expected by the reviewer, our analyses revealed that high risk constellations (GPS 2) are more frequent in neuroendocrine neoplasm of unknown primary (NNUP) and NEC compared to pan-NEN, SI-NEN and LI-NEN (page 11, line 272 and lines 281 - 284; Figure 3 K and L; Supplementary Figure 1F). Of course, we now provide related p-values for Figure 1.
  6. As suggested by the reviewer, we performed further analyses for Pan-NEN and Si NENs. In pancreatic NENs, we observed that the GPS is able to identify high risk patients (page 11, lines 279 – 281; Supplementary Figure 1 A - D). However, there was no distinction between patients with low risk (GPS0) and intermediate risk (GPS1) constellation. As the majority of SI-NET cases was associated with low-grade histology (G1/2 in 96.0% of cases (24/25) compared to 82.9% of cases (34/41) in pan-NETs), Kaplan-Meier analysis revealed that there is no clear distinction of GPS-risk groups in patients with SI-NETs. Additional survival analyses according to primary tumor sites show a significant poorer prognosis for OS in NNUP and comparable results for both PFS and OS comparing pan-NEN and SI-NEN (page 11, line 272; Figure 3L)
  7. We thank the reviewer for raising the issue concerning mutational profiles in GEP-NENs. Therefore, we edited this part in the discussion section as suggested and provided additional references (page 14, lines 380 – 382).

Reviewer 2 Report

This an important work evaluating prognostic scores in Neuroendocrine Neoplasms. The conclusions are scientifically valid.

The limitations of the rertrospective study are clearly mentioned by the authors, and further validation of the data are warranted.

Author Response

We want to thank the reviewer for the favourable evaluation of our work.

Reviewer 3 Report

The authors describe that the Glasgow-Prognostic Score predicts survival and outcomes in GEP-NEN.

 I do have certain concern and questions to your manuscript.

General comments:

1)    Pancreatic NET behave differently from Si-NET. For the Si-NET it would be interesting to now the secretion of serotonin (Hedinger-Syndrom), for Pan-Net gastrin, insulin, glucagon.

2)    Why did’t you look separately to Si-NET and Pan-NET?

3)    In 2019 there was a change in the classification of NET G3 and NEC G3. (WHO Classification of Tumours Editorial Board; Digestive System Tumours, WHO Classification of Tumours. 5th ed. IARC Press, Lyon, France2019 WHO Classification of Tumours Editorial Board; Digestive System Tumours, WHO Classification of Tumours. 5th ed. IARC PressLyon, France2019.)

If you are looking for OS, PFS you should take the change in the classification into account. You should analyse G1, G2, G3 NET and NEC G3 from SiNET and Pan-NET separately. However for a subgroup analysis our numbers are too small.

4)    Why are you taking Grad III and IV tumours into one group? A patient with a Stage IV tumour qualifized to other treatment option as a patient with a stage III.

Indroduction:

-       What’s about SIRT?

Methods:

-       Why haven’t you mentioned the serotonin, gastrin, insulin, and glucagon? It is an important marker for Si-NET in terms of treatment modalities.

Results:

-       Range of age: 15-95 or 18-5 (introduction)?

-       29 patients are not 40.8% (weight)

-       Metastatic disease:  53 or 54 patients?

-       Table 1:

o    primary site: can’t be metastasis (count to UICC IV and here the number is not equal)

o    CRP is not a title

o    Histological grading NET/NEC G3 is not the same, Ki67 is a part of the grading, you can’t put in a range

o    SSTR2 is a title

o    Aberration: you are mentioning NSE: not in the table

o    Why mention MEN? MEN 1? MEN 2, Why?

o    …..

o     

-       Table 6: 1st line treatment are more than 102!

-       The treatment options are different if you are looking at a G1 tumour UICC Stage I or a NEC G3 UICC IV. You shouldn't mix them all and give a general conclusion.

Author Response

We thank the reviewer for raising certain concerns as we now hope to present our data in a more distinct manner.

1. We totally agree with the reviewer’s concerns pointing on missing data regarding the secretion of serotonin, gastrin, insulin or glucagon. Unfortunately, in this retrospective setting, this dataset was available only in a small fraction of patients as, according to the current S2k guideline ‘Practice guideline neuroendocrine tumors’ (AWMF 2018), the measurement of hormones constitutes an optional diagnostic. We now discuss this issue in our manuscript (page 14, lines 387 – 390).

2. We are grateful for the reviewer’s suggestion of analysing Pan-NEN and Si-NEN separately. Therefore, we performed additional analyses and added our results to the current manuscript (page 11, lines 279 – 281; Supplementary Figure 1 A -D).

3. Agreeing with the reviewer’s concerns, we now take the latest version of the WHO classification into account and separated NECs from NETs (page 5, line 207; lines 211 - 212; Table 2; Figure 1; Figure 2). Moreover, we provide information regarding small cell and large cell types in NEC (Table 2). As a consequence, we performed additional statistical analyses to adequately address the issue raised here (page 9, lines 241 - 242; Table 4 and Table 5; page 11, lines 268 – 271; Figure 3; Supplementary Figure 1).

4. We are grateful for the reviewer raising an important issue here. Therefore, we renewed univariate as well as consecutive multivariate analyses counting metastatic disease (UICC IV) as factor of prognostic potential (Tables 4 and 5; page 9, lines 242 – 243; page 10, line 262).

5. Introduction: In our introduction we already discussed radioembolization (=RE or SIRT) as a therapeutic option. However, we now highlighted this therapeutic option more clearly as suggested by the reviewer (page 2, lines 89 and lines 92 – 94).

6. Methods: Please see the abovementioned point 1.

7. Results: We thank the reviewer for highlighting this mistake. The correct range is 18 – 95. We corrected the range in the abstract, the results as well as in Table 2 (page 2, line 31; page 5, line 196; Table 2). Additionally, we highlighted that information on weight disorder was available in 71 cases. Consecutively, we detected 29 weight disorders among these 71 cases (40.8%) (page 5, line 201). We now hope to present this issue more clearly. Lastly, the corrected the proportion of metastasis diseases included in the current study and aligned this information with the proportion of UICC stage IV diseases (page 1, lines 36 – 37; Table 2).

8. Following the reviewer’s instruction we renewed Table 2.

9. The explanation for the high number of first line treatments in Table 6 is that, as outlined in the legend of Table 6, we collected treatment modalities. It is possible that different modalities were combined in first line. However, we agree with the reviewer’s concerns regarding the demonstration of treatment modalities irrespective of histological grading. Consecutively, we renewed Table 6 (page 12, line 294 and lines 300 – 302; Table 6).

Reviewer 4 Report

The author of the manuscript discussed the reference to this cancer mGPS is a rare type of tumor that can form in the pancreas or in other parts of the gastrointestinal tract, including the stomach, small intestine, colon, rectum, and appendix. The GPS/mGPS is the most extensively validated of the systemic inflammation-based prognostic scores and therefore may be used in the routine clinical assessment of patients with cancer.

Minor comments:

1.    What happened to the control group of patients who did not receive GPS

2.    This is a retrospective study, and it will be more conclusive if the author add the follow-up study of patients.

3.    although the studies addressed the validity of GPS in Gastro Entero Pancreatic patients which is very rare and did not provide sufficient evidence on whether elevated GPS is prognostically efficient in all patients, i.e., with different stages of the disease and different pancreatic functional statuses,

4.    Also, the author should also mention and clarify which of the GPS (original or modified) is more suitable regarding their discriminating ability and monotonicity of gradients.

Author Response

We thank the reviewer for the appreciation of our manuscript.

1. As outlined in the Methods section, we excluded 17 patients which referred to other institutions within 30 days after initial diagnosis and another 25 patients due to subsequent loss of follow-up. Consecutively, we calculated the GPS for all GEP-NEN patients (n = 102, see also Table 2) included for further analysis.

2. We totally agree with the reviewer’s concerns regarding the retrospective nature of the current study and internally evaluated the reviewer’s suggestion. Indeed, a longer follow-up would be desirable and we already included patients over a 10-year period. We reviewed our data and had to realize that we would be able to prolong the follow-up period only in a minor subset of cases. After much discussion and minimal benefit, we decided to not include fragmentary follow-up data from the minority of patients in the cohort. Further studies are needed the validate our results. This is now addressed more clearly in the manuscript (page 15, lines 407 - 409).

3. To the greatest extent, we tried to address the reviewer’s suggestions. We performed additional analyses as outlined in Figure 1 and Supplementary Figure 1. Due to limited sample size, the additional survival analysis taking each UICC stage into account did not render any new findings (data not shown). Of course, we totally agree with the reviewer that providing this information, on the basis of a larger cohort, would be of great interest.

4. Unfortunately, we only calculated the GPS instead of the mGPS. The essential difference between both risk-scores is that hypoalbuminemia alone does not elevate the mGPS. However, we performed additional Kaplan Meier analysis that revealed non-superiority for the mGPS as the mGPS was able to identify low-risk patients more efficiently but not high-risk patients (Supplementary Figure 2). This has also been addressed in the current form of the manuscript (page 14, lines 354 - 357).

Round 2

Reviewer 1 Report

No further comments.